# Suicide among cancer patients

Nicholas G. Zaorsky [1,2], Ying Zhang[2], Leonard Tuanquin[1], Shirley M. Bluethmann[2], Henry S. Park[3] & Vernon M. Chinchilli[2]

Our purpose is to identify cancer patients at highest risk of suicide compared to the general population and other cancer patients. This is a retrospective, population-based study using nationally representative data from the Surveillance, Epidemiology, and End Results program, 1973-2014. Among 8,651,569 cancer patients, 13,311 committed suicide; the rate of suicide was 28.58/ 100,000-person years, and the standardized mortality ratio (SMR) of suicide was 4.44 (95% CI, 4.33, 4.55). The predominant patients who committed suicide were male (83%) and white (92%). Cancers of the lung, head and neck, testes, bladder, and Hodgkin lymphoma had the highest SMRs ( > 5-10) through the follow up period. Elderly, white, unmarried males with localized disease are at highest risk vs other cancer patients. Among those diagnosed at < 50 years of age, the plurality of suicides is from hematologic and testicular tumors; if > 50, from prostate, lung, and colorectal cancer patients.

[1] Department of Radiation Oncology, Penn State Cancer Institute, Hershey, PA 17033, USA. [2] Department of Public Health Sciences, Penn State College of Medicine, Hershey, PA 17033, USA. [3] Department of Therapeutic Radiology, Yale School of Medicine, Smilow Cancer Hospital at Yale, 35 Park Street, New Haven, CT 06511, USA. Correspondence and requests for materials should be addressed to N.G.Z. (email: nicholaszaorsky@gmail.com)

Cancer is the leading cause of death in the United States, and the third leading cause of death around the world[1]. In the 1900s, it was assumed that the primary goal in treating cancer was survival, sometimes at the sacrifice of physical, emotional, and financial burden. However, the import of a potentially fatal diagnosis and the long trajectory of both cancer treatment and recovery still takes a significant and sometimes overlooked toll on patients with cancer and their families. Suicide is the culmination of unmanaged distress; it is the 10th leading cause of death in the United States, and risk factors for suicide among cancer patients are similar to those among the general population, including male sex and older age[2,3]. As the survival rates of cancer patients continue to increase, it will become crucial to identify cancer patients at elevated risk of suicide.

The 2016 Joint Commission Sentinel Event Alert recommends detection of suicide risk across all health care fields[4]. Similarly, the American College of Surgeons Committee on Cancer, the American Society of Clinical Oncology, and the National Comprehensive Cancer Network have identified emotional distress and anxiety as vital signs of cancer patients[5,6]. The National Action Alliance for Suicide Prevention supports a research agenda to reduce the national suicide rate upto 20% by 2020[7]. One strategy proposed is to identify and target subgroups at greatest risk of suicide. However, clinical detection of suicide is poor;[8] moreover, there is currently no contemporary resource to assist clinicians, including oncologists and psychiatrists, in identifying cancer patients at highest risk of suicide.

The purpose of the current work is to present a contemporary analysis of the risk of suicide among cancer patients. Our objectives are to identify cancer patients at highest risk of suicide compared to (1) the general population, and (2) other cancer patients. The results of the current work suggest that suicide-prevention strategies may be aimed at those >50 years of age patients with cancer of the prostate, lung, colorectum, and bladder; as well as patients with leukemias, lymphomas, and germ cell tumors. This work may be used clinically by oncologists and psychiatrists in the creation of survivorship programs to reduce distress and anxiety and mitigate the risk of suicide among cancer patients.

## Results

A total of 8,651,569 cancer patients were included in the analysis; of these, 13,311 (0.154%) committed suicide. Among all cancer patients, the rate of suicide per 100,000-person years was 28.58, and the SMR of suicide was 4.44 (95% CI, 4.33, 4.55, $p < 0.0001$).

**Cancer patient risk of suicide vs general population.** Table 1 shows the characteristics of all cancer patients included as well as those who committed suicide vs all cancer patients. The predominant patients who committed suicide were male (11,042, 83%) and white (12,258, 92.1%). Patients who were diagnosed at a younger age had a higher SMR for suicide, and the SMRs gradually declined as patients were diagnosed at a later age: e.g. those <39-years had an SMR of 37.24 (95% CI 34.24, 40.44, $p < 0.0001$) vs>80-year-olds had an SMR of 2.40 (95% CI 2.19, 2.62, $p < 0.0001$). Although there were only 1753 (13.2%) patients with metastatic/distant disease at diagnosis, these patients had the highest SMR of death from suicide, 13.19 (95% CI 12.18, 14.26, $p < 0.0001$). There was a trend in increase in the SMR of patients who committed suicide since the 1970s through 2014: e.g. those diagnosed 1973–1980 had an SMR of 3.43 (95% CI 3.23, 3.65, $p < 0.0001$) vs those diagnosed 2011–2014 had an SMR of 36.91 (95% CI 31.91, 42.47, $p < 0.0001$).

Figure 1 shows SMRs of suicide among cancer patients by subsite. Certain cancer patients have relatively high SMR from suicide in the first year after diagnosis. For example, lung cancer patients have an SMR of 25 (95% CI 22, 28, $p < 0.0001$), and this SMR decreases to 4 after 5 years of follow-up. Similarly, Hodgkin lymphoma patients have an SMR of 26 (95% CI 13, 39, $p < 0.0001$) in the first year following diagnosis, but this SMR remains elevated throughout all follow-up times. The SMR of suicide subsides with longer follow-up time for most cancers, including colorectal, breast, bladder, head and neck, non-Hodgkin lymphoma, kidney, and endometrial. In contrast, for certain cancers, the SMR of suicide remains elevated (e.g. Hodgkin lymphoma, prostate) or increases (i.e., testicular) over follow-up time.

**Cancer patient risk of suicide vs other cancer patients.** Table 2 (left panel) shows the ORs of patients who committed suicide, stratified by subgroup. Patients older than 80 years of age have a suicide OR of 0.71 (95% CI 0.66, 0.77) compared to those <39 years of age. Males have an OR of 5.16 (95% CI 4.92, 5.40) compared to females. Blacks have an OR of 0.28 (95% CI 0.26, 0.31) compared to whites. Unmarried patients had an OR of 1.46 (95% CI 1.41, 1.52) compared to those who were married. Patients with localized disease had a higher OR of suicide compared to those with distant metastases, OR of 1.41 (95% CI 1.44, 1.63). Figure 2 shows the cancer patients who committed suicide as a function of age group. Table 2 (right panel) shows the HRs of patients who committed suicide, stratified by subgroup, complementing the results of Fig. 2. Relatively few patients <50 years of age commit suicide, in part because most cancers are diagnosed in the elderly. Among patients diagnosed at age <50, the plurality of suicide occurs in patients with leukemias and lymphomas. In contrast, among patients diagnosed at age >50, the plurality of suicides occurs in patients diagnosed with prostate, lung, and colorectal cancer. The relative risk of suicide is generally highest in older white males: HR 80 + year-olds, vs those ≤ 39 year old 2.19 (95% CI 2.01, 2.39), HR for male vs female 5.53 (95% CI 5.27, 5.80), HR for black vs white 0.31 (95% CI 0.29, 0.35).

## Discussion

We present a contemporary analysis of risk of suicide among over 8.6 million cancer patients and report that suicide risk varies as a function of disease site, age, gender, marital status, and time after diagnosis. The risk of suicide among cancer patients is four times that of the general population and has increased from the twofold risk reported in 2002[3]. The relative risk of suicide, vs the general population, is highest in those with cancer of the lung, head and neck, testes, and Hodgkin lymphoma. This relative risk of suicide decreases for most patients followed more than 5 years after diagnosis; however, risk remains elevated or rises for those with Hodgkin lymphoma and testicular cancer. The plurality of suicides occurs in adults >50 years of age with cancer of the prostate, lung, colorectum, and bladder, particularly among white, unmarried males.

Most cancer patients now die of non-cancer causes[9]. The results of the current work suggest that suicide-prevention strategies may be aimed at those >50 years of age with cancer of the prostate, lung, colorectum, and bladder; as well as patients with leukemias, lymphomas, and germ cell tumors. We recommend that providers follow the evolving guidelines for monitoring distress and suicide prevention from the American College of Surgeons Committee on Cancer, the American Society of Clinical Oncology, the National Comprehensive Cancer Network, and Action Alliance for Suicide Prevention[5–7].

**Cancer patient risk of suicide vs general population.** With respect to Objective 1, we found that although suicide contributed

**Table 1 Standardized mortality ratios of suicide among cancer patients**

Demographic summary

| | Total[a] (%) | Suicides[a] (%) | Suicides per 100,000 Person-years[a] | SMR (95% CI)[b] |
|---|---|---|---|---|
| *Age group* | | | | |
| ≤39 | 539,154 (6.2) | 898 (6.7) | 17.43 | 37.24 (34.24, 40.44) |
| 40–49 | 758,700 (8.8) | 1133 (8.5) | 19.07 | 18.05 (16.63, 19.55) |
| 50–59 | 1,551,015 (17.9) | 2221 (16.7) | 22.05 | 8.41 (7.91, 8.94) |
| 60–69 | 2,229,252 (25.8) | 3750 (28.2) | 30.11 | 4.17 (3.98, 4.38) |
| 70–79 | 2,171,816 (25.1) | 3703 (27.8) | 38.98 | 2.75 (2.62, 2.90) |
| 80+ | 1,401,632 (16.2) | 1606 (12.1) | 46.46 | 2.40 (2.19, 2.62) |
| *Sex* | | | | |
| Female | 4,210,976 (48.7) | 2269 (17.0) | 9.22 | 9.03 (8.50, 9.59) |
| Male | 4,440,593 (51.3) | 11042 (83.0) | 50.28 | 3.98 (3.87, 4.10) |
| *Race* | | | | |
| White | 7,194,990 (83.2) | 12258 (92.1) | 30.99 | 4.33 (4.21, 4.44) |
| Black | 847,121 (9.8) | 430 (3.2) | 11.28 | 4.55 (3.93, 5.24) |
| Other | 530,704 (6.1) | 532 (4.0) | 19.00 | 10.72 (9.39, 12.19) |
| Unknown | 78,754 (0.9) | 91 (0.7) | 22.09 | 0.00 (0.00, 0.00) |
| *Marital status* | | | | |
| Married | 4,788,231 (55.3) | 7371 (55.4) | 25.61 | 3.74 (3.61, 3.86) |
| Unmarried | 3,304,820 (38.2) | 4933 (37.1) | 33.13 | 6.43 (6.15, 6.71) |
| Unknown | 558,518 (6.5) | 1007 (7.6) | 34.77 | 4.43 (4.00, 4.90) |
| *Stage* | | | | |
| Distant | 1,682,112 (19.4) | 1753 (13.2) | 53.14 | 13.19 (12.18, 14.26) |
| Regional | 2,475,922 (28.6) | 4023 (30.2) | 27.69 | 5.23 (4.97, 5.49) |
| Localized | 2,800,293 (32.4) | 3999 (30.0) | 18.65 | 4.11 (3.92, 4.30) |
| Unstaged/unknown | 1,693,242 (19.6) | 3536 (26.6) | 48.41 | 3.48 (3.32, 3.64) |
| *Year of diagnosis* | | | | |
| 1973–1980 | 543,876 (6.3) | 1350 (10.1) | 30.44 | 3.43 (3.23, 3.65) |
| 1981–1990 | 905,001 (10.5) | 2511 (18.9) | 34.41 | 3.71 (3.54, 3.88) |
| 1991–2000 | 173,5021 (20.1) | 3584 (26.9) | 26.82 | 4.25 (4.05, 4.46) |
| 2001–2010 | 3,814,905 (44.1) | 4924 (37.0) | 25.73 | 8.81 (8.30, 9.35) |
| 2011–2014 | 1,652,766 (19.1) | 942 (7.1) | 40.19 | 36.91 (31.91, 42.47) |
| *Surgery* | | | | |
| Yes | 5,017,756 (58.0) | 7766 (58.3) | 21.74 | 3.98 (3.85, 4.11) |
| No | 3,399,898 (39.3) | 5106 (38.4) | 49.28 | 5.66 (5.41, 5.91) |
| Unknown | 233,915 (2.7) | 439 (3.3) | 89.49 | 4.36 (3.54, 5.32) |
| All patients | 8,651,569 | 13311 | 28.58 | 4.44 (4.33, 4.55) |

[a]Data base SEER 18 Regs Research Data + Hurricane Katrina Impacted Louisiana Cases, Nov 2016 Sub (1973–2014 varying) <Katrina/Rita Population Adjustment> was used
[b]Data base Incidence - SEER 9 Regs Research Data, Nov 2016 Sub (1973-2014) <Katrina/Rita Population Adjustment> was used; exact method was used to calculate 95% CI

to 0.154% of deaths among cancer patients, the risk of suicide is 4.4 times that of the general population (Table 1, Fig. 1), which has increased from 1.9 in patients diagnosed 1973-2002[3]. The SMR of suicide in the United States (SMRs of 9.0 for men and 4.0 for women), is higher than that in other countries. In Denmark[10], SMRs are 1.7 for men and 1.4 for women. In Norway[11], SMRs are 1.6 for men and 1.3 for women. Notably, SMRs may not be compared to one another (only to the reference population), and these differences among the countries may be due to different rates of suicide in the general populations. SMRs may be rising in the US because of the aging cancer population[12] and because most cancer patients are diagnosed with low-risk disease that is unlikely to cause cancer-specific mortality[9]. Male cancer patients are at higher risk than females, similar to the general population[13].

The SMRs are significantly higher among recently diagnosed patients (i.e., 2011–2014) compared to patients diagnosed before 2011 (Table 1). Patients diagnosed in more recent years have a shorter follow up time (i.e. until 2017) compared to those diagnosed in the 1970–2000s. Since the SMRs are generally highest in the first few years after diagnosis vs >5–10 years after diagnosis (per Fig. 1), the SMRs for the most recent patients are skewed and are higher than patients diagnosed in prior years. Notably, since SMRs are a measure to the standardized

population (the general US population in this case), SMRs should not be compared to each other, and the SMRs from Objective 1 should not be compared to the ORs in Objective 2, described below.

Additionally, we found that SMRs are highest among those diagnosed at a younger age, consistent with previous works showing that young cancer patients are a higher risk to die of any cause[9]. Notably, patients with testicular cancer have an SMR of suicide that increases over the follow-up period (SMR > 17 by 5+ years), suggesting that these patients should experience elevated distress and may benefit from close monitoring and early intervention[5,6]. These results add to the work by Fossa et al., which demonstrated that testicular cancer patients had SMRs of 1.34 for all non-cancer causes of death[14]. Similarly, patients with Hodgkin lymphoma remain at an elevated risk to die of suicide through the follow up period (SMRs 15–25, higher than most other cancers). Both subsets of cancer patients may receive systemic therapy and radiation therapy that cause infertility. Our results support the work by Kjaer et al.[15] who reported increased risk of suicide (hazard ratio, 1.68) among Danish women with fertility problems.

Finally, compared to the analysis that include patients up to 2002[3], we found that head and neck cancer patients are not the

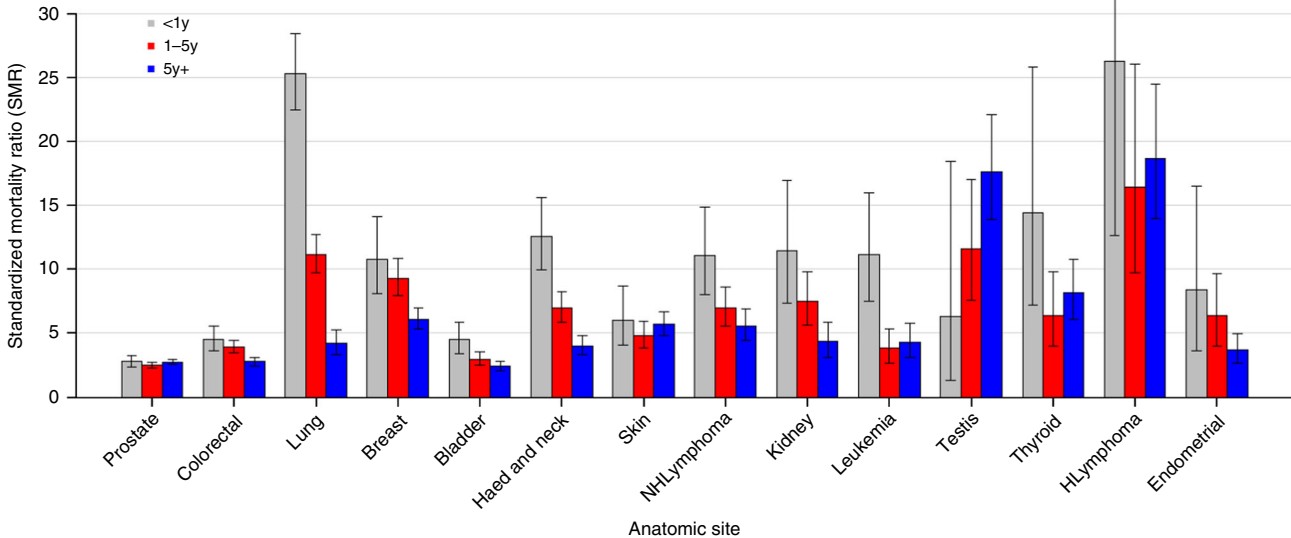

**Fig. 1** Standardized mortality ratios (SMRs) of suicide among cancer patients by subsite. The y-axis depicts the SMR with 95% CI, and the x-axis depicts the disease site. Different time periods after diagnosis (<1 year vs 1–5 years vs >5 years) are shown in different colors. Certain cancer patients have relatively high SMR from suicide in the first year after diagnosis (e.g. lung, with SMR of 25; or Hodgkin lymphoma, with SMR of 26). For most cancers, the SMR of suicide subsides with longer follow-up time. In contrast, for certain cancers, the SMR of suicide remains elevated (e.g. Hodgkin lymphoma) or increases (e.g. testicular)

**Table 2 Odds ratios and hazard ratios of suicide among cancer patients**

| | Logistic regression model | | | Cox proportional hazards model | | |
|---|---|---|---|---|---|---|
| | **Odds ratio** | **95% CI** | **P-value[a]** | **Hazard ratio** | **95% CI** | **P-value[a]** |
| *Age group* | | | <0.0001 | | | <0.0001 |
| ≤39 | – | – | | – | – | |
| 40–49 | 1.15 | (1.05, 1.25) | | 1.63 | (1.49, 1.78) | |
| 50–59 | 0.98 | (0.90, 1.06) | | 1.55 | (1.43, 1.68) | |
| 60–69 | 1.01 | (0.94, 1.09) | | 1.77 | (1.64, 1.91) | |
| 70–79 | 0.97 | (0.90, 1.05) | | 2.05 | (1.90, 2.22) | |
| 80+ | 0.71 | (0.66, 0.77) | | 2.19 | (2.01, 2.39) | |
| *Sex* | | | <0.0001 | | | <0.0001 |
| Female | – | – | | – | – | |
| Male | 5.16 | (4.92, 5.40) | | 5.53 | (5.27, 5.80) | |
| *Race* | | | <0.0001 | | | <0.0001 |
| White | – | – | | – | – | |
| Black | 0.28 | (0.26, 0.31) | | 0.31 | (0.29, 0.35) | |
| Other | 0.67 | (0.61, 0.73) | | 0.68 | (0.63, 0.75) | |
| Unknown | 0.66 | (0.53, 0.81) | | 0.61 | (0.50, 0.76) | |
| *Marital status* | | | <0.0001 | | | <0.0001 |
| Married | – | – | | – | – | |
| Unmarried | 1.46 | (1.41, 1.52) | | 1.73 | (1.67, 1.80) | |
| Unknown | 1.44 | (1.34, 1.54) | | 1.24 | (1.16, 1.33) | |
| *Stage* | | | <0.0001 | | | <0.0001 |
| Distant | – | – | | – | – | |
| Regional | 1.41 | (1.33, 1.49) | | 0.62 | (0.58, 0.66) | |
| Localized | 1.53 | (1.44, 1.63) | | 0.64 | (0.60, 0.68) | |
| Unstaged/unknown | 1.36 | (1.28, 1.44) | | 0.77 | (0.72, 0.82) | |
| *Year of diagnosis* | | | <0.0001 | | | <0.0001 |
| 1973–1980 | – | – | | – | – | |
| 1981–1990 | 1.11 | (1.04, 1.19) | | 1.01 | (0.94, 1.08) | |
| 1991–2000 | 0.82 | (0.77, 0.88) | | 0.68 | (0.64, 0.73) | |
| 2001–2010 | 0.52 | (0.48, 0.55) | | 0.58 | (0.54, 0.62) | |
| 2011–2014 | 0.23 | (0.21, 0.25) | | 0.60 | (0.55, 0.66) | |
| *Surgery* | | | 0.0001 | | | <0.0001 |
| Yes | – | – | | – | – | |
| No | 0.92 | (0.88, 0.95) | | 1.26 | (1.20, 1.31) | |
| Unknown | 0.97 | (0.87, 1.07) | | 1.03 | (0.89, 1.18) | |
| a: Type III | | | | | | |

disease site at greatest risk of suicide vs the general population. This finding may be secondary to the shifting distribution of human papilloma virus-associated cancers, and a decrease in those associated with tobacco and alcohol.

Previously, Kumar et al.[16] used the SEER database to compare suicide rates among cancer patients in the USA diagnosed in 2007–2013 vs those diagnosed in 2000–2006. The authors reported that (1) cancer patients are at a 1.37-fold higher risk

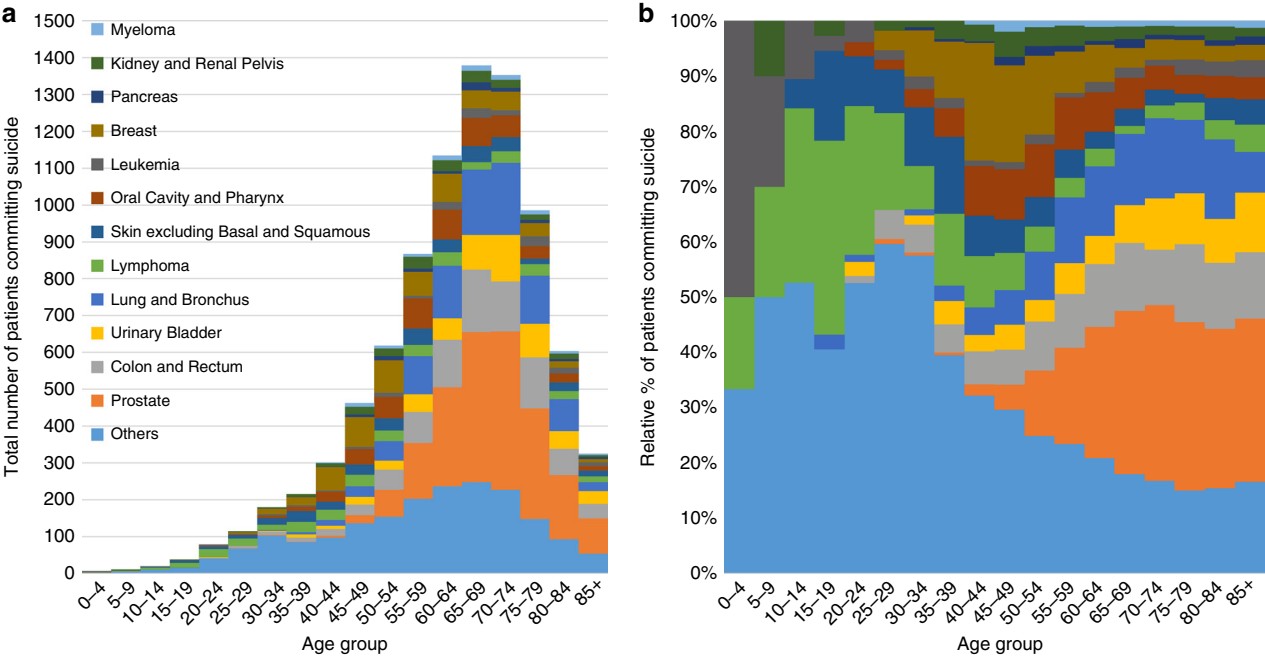

**Fig. 2** Suicide among cancer patients as a function of age group. **a** The y-axis depicts the absolute number of suicides and the x-axis depicts the age group at time of diagnosis. The colors depict the disease sites. The majority of suicides are in patients diagnosed at an older age (i.e. 50–80-year-olds), and the plurality of suicides s occurs in patients diagnosed with prostate, lung, and colorectal cancer. **b** The y-axis depicts the relative number of suicides compared to all cancer patients, and the x-axis depicts the age group at time of diagnosis. The colors depict the disease sites. For younger patients (i.e. <50), the plurality of suicides is seen in lymphoma patients. In contrast, among older adults (i.e. >50) the plurality of suicides occurs in patients with cancer of the prostate, lung, colorectum, and bladder

of committing suicide vs the general population; (2) the risk is highest in older males, in the first year of diagnosis; and (3) the rate of suicide has not increased between the two time periods included (i.e. 2007–2013 vs 2000–2006).

Our current work adds to the work by Kumar et al. First, we characterize suicide rates from 1973 to 2015, and we find that the OR of suicide is decreasing in more recent years vs previous years (i.e. 2011–2014 vs 1973–1980), however the risk of suicide vs. the general population is increasing: 1.9 in patients diagnosed 1973–2002[3], vs. 4.4 for patients in the current work who were diagnosed 1973–2015. Additionally, we compare the relative risk of death from suicide vs the general population (in Objective 1), as well as vs other cancer patients (in Objective 2). Further, we identify distinct subgroups of cancer patients who contribute to the plurality of suicides, i.e. those >50 years of age with cancer of the prostate, lung, colorectum, and bladder; as well as patients with leukemias, lymphomas, and germ cell tumors. The results of the current study may be used to guide interventions for suicide prevention among unique subgroups.

**Cancer patient risk of suicide vs other cancer patients**. With respect to Objective 2, we found that the plurality of suicides occurs adults >50 years of age with cancer of the prostate, lung, colorectum, and bladder (Table 2, Fig. 2). Cancer is typically diagnosed among the elderly, and these cancers are not prevalent among younger patients; in contrast, leukemias, lymphomas, and testicular cancer are more common among younger patients, and adolescents who are subsequently diagnosis have an elevated risk of suicide.

We found that men have an OR of 5.16 in committing suicide, which corroborates in our findings from Objective 1, and is also likely secondary to the incidence of testicular cancer being only in male patients. Additionally, patients who were unmarried had

65% of the odds of committing suicide, those who were white had almost 400% as those who were black. The odds of suicide were highest among those diagnosed in more distant decades, likely because of the increase time at risk, particularly among cancer survivors.

The OR of suicide of patients diagnosed in more recent years is lower than that diagnosed in previous years (e.g. 2011–2014 vs 1973–1980 in Table 2), suggesting that patients diagnosed in more recent years are less likely to commit suicide than patients diagnosed in previous years. This finding is likely secondary to the evolving characteristics of cancer patients in the USA; with a decrease in smoking rates (highest among elderly white males), there is a decrease in rates of lung cancer and human papilloma virus (HPV)-negative head and neck cancers, which have also historically been cancers of elderly white men. In contrast, with the advent of screening mammography and prostate specific antigen (PSA) testing in the 1990s, there has been a surge in the diagnoses of low-risk breast and prostate cancers[1]. ORs compare the odds of suicide of the group of one group of cancer patients vs a reference group of cancer patients, unlike the SMRs in Objective 1, which compare relative risk of death vs. the entire US population, as a function of time after diagnosis. Thus, if there is a change in the rate of suicide in subpopulations of patients between two eras, this change will be reflected in the ORs, but not necessarily in the SMRs.

Our work has limitations. The overall number of deaths from suicide was relatively limited overall (<1% of cancer patients), and more detailed analyses on risk factors could not be performed. Treatment paradigms have changed since the 1970s; for example, Hodgkin lymphoma patients are now treated with limited chemotherapy, and possibly a relatively low dose of very targeted radiation[17]. Additionally, patients having death events in earlier years (i.e. 1973–1983) have limited follow up and less time at risk (≤10 years) than some patients with events in more recent years.

This may have resulted in an overestimate of SMRs for individuals diagnosed between 2011–2014, compared to those diagnosed before 1980. Similarly, patients diagnosed in recent years have short follow-up and lower chance of death from any cause.

Further, there is a risk of bias and misclassification of suicide in the SEER[18,19]. For example, in a review of cases of non-Hodgkin lymphomas, agreement in the subclassification of histologies between the expert review and the SEER registry record varied from 5 to 100%[20]. An investigation of the California Cancer Registry, which contributes to the national SEER data[21], revealed registry sensitivity of receipt of radiation therapy of only 72%. As of 2018, there has been limited research published regarding the misclassification of cause of death in the SEER database. Thus, we are unable to characterize misclassification of suicide in the current work.

Nonetheless, for suicide, there is likely little discrepancy in the cause of death, as compared to a cause of death like heart disease, which may be cause by the cancer treatment, underlying heart disease, or a combination of both. We agree with Sun and Trinh, in their assessment of the SEER database, that although there may be some errors in large registries, the errors are likely less frequent than those in hospital based databases and big data will continue to remain an integral part of hypothesis-generating exploratory analyses in medical research[19].

The results of the current work suggest that suicide-prevention strategies may be aimed at those >50 years of age patients with cancer of the prostate, lung, colorectum, and bladder; as well as patients with leukemias, lymphomas, and germ cell tumors. We recommend that providers follow the evolving guidelines for monitoring distress and suicide prevention from the American College of Surgeons Committee on Cancer, the American Society of Clinical Oncology, the National Comprehensive Cancer Network, and Action Alliance for Suicide Prevention[5–7].

## Methods

**Data acquisition.** Patients with invasive cancer, diagnosed between 1973 and 2014, were abstracted from the National Cancer Institute's Surveillance, Epidemiology, and End Results (SEER) program[22,23]. The overview and limitations of the database and the methods are described in the Supplementary information[24–26]. SEER is a network of population-based incident tumor registries from geographically distinct regions in the US, covering 28% of the US population, including incidence, survival, and treatment (e.g. radiation therapy, surgery, and chemotherapy)[22,23]. For the current analysis, the SEER18 registry was used. The SEER registry includes data on sex, age at diagnosis, race, marital status, and year of diagnosis. SEER does not code comorbidities, performance status, surgical pathology, margin status, doses, or chemotherapy agents. SEER*Stat 8.2.1 was used for analysis[22].

All patients with an invasive cancer diagnosis were included. Patients diagnosed only through autopsy or death certificate (<1.5% of patients) were excluded. Data were extracted for patients with more than 100,000 person-years or more of survival time; thus, certain uncommon and aggressive cancers were excluded, including Kaposi's sarcoma, multiple myeloma, hepatobiliary cancer, male breast cancers, and mesotheliomas. Since certain cancers represent a heterogeneous group of disease (e.g. leukemia, lymphoma), these cancers were grouped for certain parts of the analysis, so they could be reported accurately.

Mortality codes in SEER are assigned from death certificates, completed by the doctor caring for the patient at the time of demise. Patients were considered to have committed suicide if the cause of death was coded as: suicide and self-inflicted injury (50220). Patients with other causes of death, including accident, homicide, and legal intervention were excluded from the present analysis. Survival time in SEER is measured in months, and the smallest nonzero value is 1 month, which was the minimum time to any event.

Notably, SEER does not code comorbidities or diagnoses associated with suicide, including suicidal ideation, previous suicide attempts, or use of anti-depressive medications. The observed associations between cancer and suicide may be confounded by psychiatric disorders and the use of medications, but we are unable to control for these factors in the current work. These are limitations of the analysis and limit the interpretability of the results.

For objective 1, we calculated standardized mortality ratios (SMRs), which provide the relative risk of death for patients with cancer as compared to all US residents, stratified by cancer subgroup[3,22,27]. Data were characterized with SMRs adjusted by age, race, and sex to the US population over the same time. Five-year age categories were used for standardization using SEER*Stat 8.2.1 and Microsoft Excel 15.0.4 (Microsoft, Redmond, WA)[27–29].

For objective 2, odds ratios (ORs) with 95% CIs were calculated based on the number of observed events per patient subgroup. Further, the absolute and relative number of suicides per patient age group (at time of diagnosis) were calculated. We also performed a survival analysis using a Cox proportional hazards model to calculate hazard ratios (HRs), with the survival time being from diagnosis until suicide, and non-suicide deaths plus living patients being censored.

**Reporting Summary.** Further information on experimental design is available in the Nature Research Reporting Summary linked to this article.

**Data availability.** The data are provided in the SEER database, which is freely accessible to the public. The relevant session information, i.e. the user-submitted request, from in the current work and abbreviated data set (from SEER) are provided in Supplementary Data Sets 1-3.

We comply with all relevant ethical regulations. The datasets generated and analyzed during the current study are available in the SEER repository (https://seer.cancer.gov/seerstat/). These data are freely available via the National Cancer Institute SEER program, and thus the study was exempt from institutional review board review. There are no participants in the study, and thus there is no consent form. To access the data in this study, we provide further detailed instruction in the Supplementary Methods.

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

## Author contributions

All authors had full access to all of the data in the study and take responsibility for the integrity of the data and the accuracy of the data analysis. Study concept and design: N.Z., Y.Z., and V.C.; Acquisition, analysis, and interpretation of data: N.Z., Y.Z., and V.C.; Drafting of the manuscript: N.Z.; Critical revision of the manuscript for important intellectual content: all authors. Statistical analysis: N.Z., Y.Z., and V.C.; Obtained funding: NA. Administrative, technical, or material support: N.Z. Study supervision: N.Z. and V.C.

## Additional information

**Competing interests:** The authors declare no competing interests.

