## [Peer Review File · Nature Communications]

Reviewers' Comments:

Reviewer #1:

Remarks to the Author:

This well-written and succinct manuscript described rates of suicide among cancer patients. The manuscript is characterized by several strengths, including an important topic, large sample size, and appropriate statistical analyses. The manuscript adds new knowledge to existing literature on this topic. The manuscript could be strengthened by addressing two points. First, SMR is significantly higher among recently diagnosed patients (i.e., 2011-2014) compared to patients diagnosed before 2011 (Table 1). Nevertheless, the odds ratio for suicide among more recently diagnosed patients is significantly lower than patients diagnosed before 2011 (Table 2). It would be helpful if the discussion could more directly explain these seemingly discrepant results. Second, it would be helpful to place results in the context of national requirements (or lack thereof) of screening for suicidality in cancer patients.

Reviewer #2:

Remarks to the Author:

The manuscript examined the suicide rate among patients with cancer from SEER database, and found that suicide rate was higher in patients with cancer compared with that in the general population. Some factors associated with suicide among patients with cancer were also studied. The topic in this manuscript were investigated previously in both US and other countries, and the findings were similar to previous reports. See the most recent publication using the same SEER database as the authors used in this manuscript (Kumar V, et al. 2017 American J Psychiatry). The manuscript, however, is a bit too compact (only ~2000 words) and lacks details of the rationale, methods, results, discussion of the implications, etc.

For instance, in the Abstract, the authors stated that "The exposure was suicide, ..." , which is not right. Suicide is the main outcome, not the exposure.

The rationale of using logistic regression model to examine the risk factors of suicide should also be clarified. The time from diagnosis to suicide could be more interesting.

The conclusions state "We recommend clinics screen older patients with cancer of Patients with should be screened indefinitely" lack empirical support or specific practical strategies from this study.

In this study, suicide was defined according to the death certificate. Misclassification of suicide is thus possible and should be discussed. Suicidal ideation should definitely be analyzed as well if it is available in the SEER database.

The observed association between cancer and suicide could be confounded by psychiatric disorders (or the like), the authors did not comment on confounding, confounding control, etc. in the study design or analysis.

Other limitations include the unclear logic between the rationale and purpose in the Introduction section, the missing flow-chart of SEER data manipulation, etc.

Dear Editors,

Enclosed, please find our revised manuscript titled, “**Suicide among cancer patients**,” for consideration in *Nature Communications*. In the manuscript, we have addressed all of the points brought up by our reviewers. In this letter, we explain how we address the reviewers’ concerns.

Additionally, we have reviewed and completed the journal manuscript checklist, the reporting summary, and the editorial policy checklist. We provide these files with the upload.

Referees’ Comments:

Reviewer #1 (Remarks to the Author):

This well-written and succinct manuscript described rates of suicide among cancer patients. The manuscript is characterized by several strengths, including an important topic, large sample size, and appropriate statistical analyses. The manuscript adds new knowledge to existing literature on this topic.

AUTHOR RESPONSE: Thank you for your comments.

The manuscript could be strengthened by addressing two points. First, SMR is significantly higher among recently diagnosed patients (i.e., 2011-2014) compared to patients diagnosed before 2011 (Table 1). Nevertheless, the odds ratio for suicide among more recently diagnosed patients is significantly lower than patients diagnosed before 2011 (Table 2). It would be helpful if the discussion could more directly explain these seemingly discrepant results.

AUTHOR RESPONSE: Thank you for your valuable suggestions. In the discussion section, we have made the following changes:

In the discussion regarding Objective 1, which uses SMRs, we state:

The SMRs are significantly higher among recently diagnosed patients (i.e., 2011-2014) compared to patients diagnosed before 2011 (Table 1). Patients diagnosed in more recent years have a shorter follow up time (i.e. until 2017) compared to those diagnosed in the 1970s-2000s. Since the SMRs are generally highest in the first few years after diagnosis vs > 5-10 years after diagnosis (per Figure 1), the SMRs for the most recent patients are skewed and are higher than patients diagnosed in prior years. Notably, since SMRs are a measure to the standardized population (the general US population in this case), SMRs should not be compared to each other, and the SMRs from Objective 1 should not be compared to the ORs in Objective 2, described below.

In the discussion regarding Objective 2, which uses ORs, we state:

The OR of suicide of patients diagnosed in more recent years is lower than that diagnosed in previous years (e.g. 2011-2014 vs 1973-1980 in Table 2), suggesting that patients diagnosed in more recent years are less likely to commit suicide than patients diagnosed in previous years. This finding is likely secondary to the evolving characteristics of cancer patients in the USA; with a decrease in smoking rates (highest among elderly white males), there is a decrease in rates of lung cancer and human papilloma virus (HPV)-negative head and neck cancers, which have also historically been cancers of elderly white men. In contrast, with the advent of screening mammography and prostate specific antigen (PSA) testing in the 1990s, there has been a surge in the diagnoses of low-risk breast and prostate cancers. ORs compare the

odds of suicide of the group of one group of cancer patients vs a reference group of cancer patients, unlike the SMRs in Objective 1, which compare relative risk of death vs. the entire US population, as a function of time after diagnosis. Thus, if there is a change in the rate of suicide in subpopulations of patients between two eras, this change will be reflected in the ORs, but not necessarily in the SMRs.

Second, it would be helpful to place results in the context of national requirements (or lack thereof) of screening for suicidality in cancer patients.

AUTHOR RESPONSE: Thank you for your valuable comments. We discuss the available national resources for screening for suicidality in cancer patients. We have modified our recommendations for screening per the comments of Reviewer #2. We now state:

The results of the current work suggest that suicide-prevention strategies may be aimed at those >50 years of age patients with cancer of the prostate, lung, colorectum, and bladder; as well as patients with leukemias, lymphomas, and germ cell tumors. We recommend that providers follow the evolving guidelines for monitoring distress and suicide prevention from the American College of Surgeons Committee on Cancer, the American Society of Clinical Oncology, the National Comprehensive Cancer Network, and Action Alliance for Suicide Prevention

Reviewer #2 (Remarks to the Author):

The manuscript examined the suicide rate among patients with cancer from SEER database, and found that suicide rate was higher in patients with cancer compared with that in the general population. Some factors associated with suicide among patients with cancer were also studied. The topic in this manuscript were investigated previously in both US and other countries, and the findings were similar to previous reports. See the most recent publication using the same SEER database as the authors used in this manuscript (Kumar V, et al. 2017 American J Psychiatry). The manuscript, however, is a bit too compact (only ~2000 words) and lacks details of the rationale, methods, results, discussion of the implications, etc.

For instance, in the Abstract, the authors stated that "The exposure was suicide, ..." , which is not right. Suicide is the main outcome, not the exposure.

AUTHOR RESPONSE: Thank you for your valuable comment. We have corrected this typo.

The rationale of using logistic regression model to examine the risk factors of suicide should also be clarified. The time from diagnosis to suicide could be more interesting.

AUTHOR RESPONSE: Thank you for your valuable comment. We applied the logistic regression analyses to report odds ratios, and we did not perform a time-to-event analysis (such as a Cox regression model). In the manuscript, we now state:

The choice of modeling depends on the study objectives and interests. In objective 2, we explored the relation between risk factors for patients committing suicide vs not committing suicide. We chose logistic regression because we were interested in whether the patients committed suicide or not, eventually, without a time factor until suicide. We did not perform a time-to-event analysis, which aims to explore the relation between risk factors and time to suicide, which was not our focus in the current work. Moreover, to fit a logistic regression model for a time-to-event analysis, we would need more events. In the current data set, we have relatively few events, 13311/8651569 (~0.15%), and the remaining patients are censored (we were not able to observe their events), which would decrease the reliability of a logistic regression model.

We value your statistical insight, and we have clarified the methods section accordingly.

The conclusions state "We recommend clinics screen older patients with cancer of Patients with should be screened indefinitely" lack empirical support or specific practical strategies from this study.

AUTHOR RESPONSE: Thank you for your valuable comment. We have revised this statement. We instead now summarize our findings and reference the evolving recommendations from national organizations. We now state:

The results of the current work suggest that suicide-prevention strategies may be aimed at those >50 years of age patients with cancer of the prostate, lung, colorectum, and bladder; as well as patients with leukemias, lymphomas, and germ cell tumors. We recommend that providers follow the evolving guidelines for monitoring distress and suicide prevention from the American College of Surgeons Committee on Cancer, the American Society of Clinical Oncology, the National Comprehensive Cancer Network, and Action Alliance for Suicide Prevention.

In this study, suicide was defined according to the death certificate. Misclassification of suicide is thus possible and should be discussed. Suicidal ideation should definitely be analyzed as well if it is available in the SEER database.

The observed association between cancer and suicide could be confounded by psychiatric disorders (or the like), the authors did not comment on confounding, confounding control, etc. in the study design or analysis.

AUTHOR RESPONSE: Thank you for your valuable comments. We have revised the methods and discussion sections as follows:

In the discussion we state:

Further, there is a risk of bias and misclassification of suicide in the SEER. For example, in a review of cases of non-Hodgkin lymphomas, agreement in the subclassification of histologies between the expert review and the SEER registry record varied from 5% to 100%. An investigation of the California Cancer Registry, which contributes to the national SEER data revealed registry sensitivity of receipt of radiation therapy of only 72%. As of 2018, there has been limited research published regarding the misclassification of cause of death in the SEER database. Thus, we are unable to characterize misclassification of suicide in the current work.

Nonetheless, for suicide, there is likely little discrepancy in the cause of death, as compared to a cause of death like heart disease, which may be cause by the cancer treatment, underlying heart disease, or a combination of both. We agree with Sun and Trinh, in their assessment of the SEER database, that although there may be some errors in large registries, the errors are likely less frequent than those in hospital based databases and big data will continue to remain an integral part of hypothesis-generating exploratory analyses in medical research.

In the methods we state:

Notably, SEER does not code comorbidities or diagnoses associated with suicide, including suicidal ideation, previous suicide attempts, or use of anti-depressive medications. The observed associations between cancer and suicide may be confounded by psychiatric disorders and the use of medications, but we are unable to control for these factors in the current work. These are limitations of the analysis and limit the interpretability of the results.

Other limitations include the unclear logic between the rationale and purpose in the Introduction section, the missing flow-chart of SEER data manipulation, etc.

AUTHOR RESPONSE: Thank you for your valuable comments. We have revised the introduction section. Regarding the introduction, we now state:

Cancer is the leading cause of death in the United States, and the third leading cause of death around the world. In the 1900s, it was assumed that the primary goal in treating cancer was survival, sometimes at the sacrifice of physical, emotional, and financial burden. However, the import of a potentially fatal diagnosis and the long trajectory of both cancer treatment and recovery still takes a significant and sometimes overlooked toll on patients with cancer and their families. Suicide is the culmination of unmanaged distress; it is the 10th leading cause of death in the United States, and risk factors for suicide among cancer patients are similar to those among the general population, including male sex and older age. As the survival rates of cancer patients continue to increase, it will become crucial to identify cancer patients at elevated risk of suicide.

Regarding the data manipulation, we now provide the Session Information from SEER and the dataset in the Supplementary Materials section. The first sheet of the XLS files contain the "session information," i.e. how we extracted the data from SEER. The subsequent sheets in each file contain the raw data.

Thank you again for your consideration. We look forward to your response.

Nicholas G Zaorsky, MD
Tenure-track Assistant Professor, Department of Radiation Oncology, Penn State Cancer Institute
Assistant Professor, Department of Public Health Sciences, Penn State College of Medicine
Hershey, PA 17033
USA. Tel: +1-717-531-8024
Fax: +1-717-531-0446
E-mail: nicholaszaorsky@gmail.com

Vernon M Chinchilli, PhD
Distinguished Professor and Chair
Department of Public Health Sciences, Penn State College of Medicine
90 Hope Drive, Suite 2200
Hershey, PA 17033-0855
Phone: 717-531-4262
FAX: 717-531-4359
Email: vchinch@psu.edu

Reviewers' Comments:

Reviewer #1:

Remarks to the Author:

The authors have done a nice job addressing reviewers' comments; the manuscript has been significantly strengthened as a result.

Reviewer #2:

Remarks to the Author:

The authors addressed some of the comments. However, the following concerns still need clarification.

1. The previous publication by Kumar V, et al. 2017 American J Psychiatry used the same SEER database with updated information until 2013 and found similar results. The present study used data until 2014? What is the novel knowledge the authors would like to contribute?

2. The authors argued that they were only interested in whether or not suicide would occur for patients with cancer. As shown in the discussion section, the authors have made suggestions for intervention for these patients with cancer. It is therefore very important in the timing of the intervention. Previous studies have found that suicide rates among patients with cancer were higher right after cancer diagnosis and then declined later. The trajectories of suicide rates are thus of significance for both research and intervention. The authors argued that "we have relatively few events, 13311/8651569 (~0.15%), and the remaining patients are censored (we were not able to observe their events), which would decrease the reliability of a logistic regression model". Since the reliability of the "logistic regression model" would be decreased, why did you use it? The survival analysis approach does have the concern about the number of events. The current data have 13311 events, which is sufficient for the survival model construction. Analyzing the data using survival analysis approaches is highly recommended.

Reviewers' comments:

Reviewer #1 (Remarks to the Author):

The authors have done a nice job addressing reviewers' comments; the manuscript has been significantly strengthened as a result.

AUTHOR RESPONSE: Thank you for your comments.

Reviewer #2 (Remarks to the Author):

The authors addressed some of the comments. However, the following concerns still need clarification.

1. The previous publication by Kumar V, et al. 2017 American J Psychiatry used the same SEER database with updated information until 2013 and found similar results. The present study used data until 2014? What is the novel knowledge the authors would like to contribute?

AUTHOR RESPONSE: Thank you for your helpful comments. In the discussion, we now state:

Previously, Kumar et al. used the SEER database to compare suicide rates among cancer patients in the USA diagnosed in 2007-2013 vs those diagnosed in 2000-2006. The authors reported that (1) cancer patients are at a 1.37-fold higher risk of committing suicide vs the general population; (2) the risk is highest in older males, in the first year of diagnosis; and (3) the rate of suicide has not increased between the two time periods included (i.e. 2007-2013 vs 2000-2006).

Our current work adds to the work by Kumar et al. First, we characterize suicide rates from 1973 to 2015, and we find that the OR of suicide is decreasing in more recent years vs previous years (i.e. 2011-2014 vs 1973-1980), however the risk of suicide vs. the general population is increasing: 1.9 in patients diagnosed 1973-2002, vs. 4.4 for patients in the current work who were diagnosed 1973-2015. Additionally, we compare the relative risk of death from suicide vs the general population (in Objective 1), as well as vs other cancer patients (in Objective 2). Further, we identify distinct subgroups of cancer patients who contribute to the plurality of suicides, i.e. those >50 years of age with cancer of the prostate, lung, colorectum, and bladder; as well as patients with leukemias, lymphomas, and germ cell tumors. The results of the current study may be used to guide interventions for suicide prevention among unique subgroups.

2. The authors argued that they were only interested in whether or not suicide would occur for patients with cancer. As shown in the discussion session, the authors have made suggestions for intervention for these patients with cancer. It is therefore very important in the timing of the intervention. Previous studies have found that suicide rates among patients with cancer were higher right after cancer diagnosis and then declined later. The trajectories of suicide rates are thus of significance for both research and intervention. The authors argued that "we have relatively few events, 13311/8651569 (~0.15%), and the remaining patients are censored (we were not able to observe their events), which would decrease the reliability of a logistic regression model". Since the reliability of the "logistic regression model" would be decreased, why did you use it? The survival analysis approach does have the concern about the number of events. The current data have 13311 events, which is sufficient for the survival model construction. Analyzing the data using survival analysis approaches is highly recommended.

AUTHOR RESPONSE: Thank you for your helpful comments. We have performed a survival analysis, as you recommend.

In the methods section, we now state:

For objective 2, odds ratios (ORs) with 95% CIs were calculated based on the number of observed events per patient subgroup. Further, the absolute and relative number of suicides per patient age group (at time of diagnosis) were calculated. We also performed a survival analysis using a Cox proportional hazards model to calculate hazard ratios (HRs), with the survival time being time from diagnosis until suicide, and non-suicide deaths or alive patients being censored.

In the results, we now state:

Figure 2 shows the cancer patients who committed suicide as a function of age group. Table 2 (right panel) shows the HRs of patients who committed suicide, stratified by subgroup, complementing the results of Figure 2. Relatively few patients < 50 years of age commit suicide, in part because most cancers are diagnosed in the elderly. Among patients diagnosed at age <50, the plurality of suicide occurs in patients with leukemias and lymphomas. In contrast, among patients diagnosed at age > 50, the plurality of suicides occurs in patients diagnosed with prostate, lung, and colorectal cancer. The relative risk of suicide is generally highest in older white males: HR 80+ year-olds, vs those ≤ 39 year old 2.19 (95% CI 2.01, 2.39), HR for male vs female 5.53 (95% CI 5.27, 5.80), HR for black vs white 0.31 (95% CI 0.29, 0.35).

We juxtapose the results of the logistic regression model (odds ratios) to the Cox proportional hazards model (hazard ratios) in Table 2:

	Logistic Regression Model			Cox Proportional hazards model		
	Odds Ratio	95% CI	P-value ^a	Hazard Ratio	95% CI	P-value ^a
Age Group			<.0001			<.0001
<=39	-	-		-	-	
40-49	1.15	(1.05, 1.25)		1.63	(1.49, 1.78)	
50-59	0.98	(0.90, 1.06)		1.55	(1.43, 1.68)	
60-69	1.01	(0.94, 1.09)		1.77	(1.64, 1.91)	
70-79	0.97	(0.90, 1.05)		2.05	(1.90, 2.22)	
80+	0.71	(0.66, 0.77)		2.19	(2.01, 2.39)	
Sex			<.0001			<.0001
Female	-	-		-	-	
Male	5.16	(4.92, 5.40)		5.53	(5.27, 5.80)	
Race			<.0001			<.0001
White	-	-		-	-	
Black	0.28	(0.26, 0.31)		0.31	(0.29, 0.35)	
Other	0.67	(0.61, 0.73)		0.68	(0.63, 0.75)	
Unknown	0.66	(0.53, 0.81)		0.61	(0.50, 0.76)	
Marital Status			<.0001			<.0001
Married	-	-		-	-	
Unmarried	1.46	(1.41, 1.52)		1.73	(1.67, 1.80)	
Unknown	1.44	(1.34, 1.54)		1.24	(1.16, 1.33)	
Stage			<.0001			<.0001
Distant	-	-		-	-	
Regional	1.41	(1.33, 1.49)		0.62	(0.58, 0.66)	
Localized	1.53	(1.44, 1.63)		0.64	(0.60, 0.68)	
Unstaged/unknown	1.36	(1.28, 1.44)		0.77	(0.72, 0.82)	
Year of Diagnosis			<.0001			<.0001
1973-1980	-	-		-	-	

1981-1990	1.11	(1.04, 1.19)	1.01	(0.94, 1.08)
1991-2000	0.82	(0.77, 0.88)	0.68	(0.64, 0.73)
2001-2010	0.52	(0.48, 0.55)	0.58	(0.54, 0.62)
2011-2014	0.23	(0.21, 0.25)	0.60	(0.55, 0.66)
Surgery			0.0001	<.0001
Yes	-	-	-	-
No	0.92	(0.88, 0.95)	1.26	(1.20, 1.31)
Unknown	0.97	(0.87, 1.07)	1.03	(0.89, 1.18)
a: Type III				

Thank you again for your consideration. We believe we have addressed all of the concerns of our reviewers. We look forward to your response.

Sincerely,

Nicholas G Zaorsky, MD
Tenure-track Assistant Professor, Department of Radiation Oncology, Penn State Cancer Institute
Assistant Professor, Department of Public Health Sciences, Penn State College of Medicine
Hershey, PA 17033
USA. Tel: +1-717-531-8024
Fax: +1-717-531-0446
E-mail: nicholaszaorsky@gmail.com

Vernon M Chinchilli, PhD
Distinguished Professor and Chair
Department of Public Health Sciences, Penn State College of Medicine
90 Hope Drive, Suite 2200
Hershey, PA 17033-0855
Phone: 717-531-4262
FAX: 717-531-4359
Email: vchinch@psu.edu

Reviewers' Comments:

Reviewer #2:

Remarks to the Author:

I have no further comments.